# Interaction proteome of human Hippo signaling: modular control of the co-activator YAP1

Simon Hauri[1,2], Alexander Wepf[3], Audrey van Drogen[1], Markku Varjosalo[1,4], Nic Tapon[5], Ruedi Aebersold[1,2,6] & Matthias Gstaiger[1,2,*]

## Abstract

**Tissue homeostasis is controlled by signaling systems that coordinate cell proliferation, cell growth and cell shape upon changes in the cellular environment. Deregulation of these processes is associated with human cancer and can occur at multiple levels of the underlying signaling systems. To gain an integrated view on signaling modules controlling tissue growth, we analyzed the interaction proteome of the human Hippo pathway, an established growth regulatory signaling system. The resulting high-resolution network model of 480 protein-protein interactions among 270 network components suggests participation of Hippo pathway components in three distinct modules that all converge on the transcriptional co-activator YAP1. One of the modules corresponds to the canonical Hippo kinase cassette whereas the other two both contain Hippo components in complexes with cell polarity proteins. Quantitative proteomic data suggests that complex formation with cell polarity proteins is dynamic and depends on the integrity of cell-cell contacts. Collectively, our systematic analysis greatly enhances our insights into the biochemical landscape underlying human Hippo signaling and emphasizes multifaceted roles of cell polarity complexes in Hippo-mediated tissue growth control.**

**Keywords** AP-MS; cell polarity; Hippo signaling; modularity; protein complex analysis

**Subject Categories** Network Biology; Signal Transduction

**Mol Syst Biol. 9: 713**

## Introduction

Development of metazoan tissues and organs depends on tight control of proliferation, cell growth and programmed cell death in response to extracellular and intracellular signals. Genetic and biochemical experiments originally performed in *Drosophila melanogaster* led to the discovery of the Hippo (Hpo) pathway, a conserved signaling cascade that controls tissue and organ homeostasis in metazoans. Its core components are conserved in humans and have been implicated in a variety of human cancers (Pan, 2010; Zhao *et al*, 2010; Harvey *et al*, 2013). These include the STE20 kinases MST1 and MST2 (orthologs of the *Drosophila* Hpo kinase) which bind to SAV1 (WW45), the AGC kinase LATS1 (Large tumor suppressor homolog 1) and its associated scaffold proteins MOB1A/B (Harvey & Tapon, 2007; Pan, 2007). Downstream of this kinase cascade are the WW-domain-containing transcriptional co-activators YAP1 and TAZ (Dong *et al*, 2007; Zhao *et al*, 2007; Lei *et al*, 2008) the two major effectors of the Hpo pathway. Active MST1/2 in complex with SAV1 phosphorylates LATS1/2, which in turn stimulates LATS-MOB complex formation and activation of LATS kinase activity. Active LATS1/2 kinases phosphorylate and inactivate YAP1 and TAZ through 14-3-3 protein-mediated cytoplasmic sequestration (Dong *et al*, 2007; Guo *et al*, 2007; Zhao *et al*, 2007; Lei *et al*, 2008; Oka *et al*, 2008; Zhang *et al*, 2008). When the Hpo pathway is inactive, hypo-phosphorylated nuclear YAP1 and TAZ bind to the TEA domain transcription factors (TEAD1/2/3/4) to drive expression of pro-growth and anti-apoptotic genes (Wu *et al*, 2008; Zhao *et al*, 2008).

While the signaling mechanism for the Hpo core module is well-established, our understanding of the physiological cues, signaling components and mechanisms that control the activation and repression of the human Hpo pathway is still quite limited. Recent genetic data from *Drosophila* and biochemical analysis in human cells suggest that Hpo signaling is linked to cell polarity, the cytoskeleton and cell junctions (Genevet & Tapon, 2011; Schroeder & Halder, 2012). However, the molecular complexes that transmit polarity and cytoskeletal signals to the Hpo core modules are just beginning to emerge and there is debate as to whether these are dependent on the core cascade or if they act directly on YAP1.

Since most proteins exert their function in the context of specific protein complexes, the characterization of complexes involving genetically defined Hpo components turned out to be a particularly successful approach to uncover novel regulators and mechanisms

1   Institute of Molecular Systems Biology, ETH Zürich, Zürich, Switzerland
2   Competence Center for Systems Physiology and Metabolic Diseases, ETH Zürich, Zürich, Switzerland
3   Analytica Medizinische Laboratorien AG, Zurich, Switzerland
4   Institute of Biotechnology, University of Helsinki, Helsinki, Finland
5   Cancer Research UK, London Research Institute, London, UK
6   Faculty of Science, University of Zürich, Zürich, Switzerland
    *Corresponding author. Tel: +41 44 633 71 49; Fax: +41 44 633 10 51; E-mail: gstaiger@imsb.biol.ethz.ch

underlying the control of tissue growth by the Hpo signaling network. Affinity purification coupled to mass spectrometry (AP-MS) has proven to be a sensitive tool for the identification of novel protein interactions under physiological conditions (Rigaut et al, 1999; Gingras et al, 2007; Gstaiger & Aebersold, 2009; Pardo & Choudhary, 2012). Using a combined AP-MS and genomics approach we recently identified dSTRIPAK, a serine/threonine phosphatase complex that associates with Hpo kinase and negatively regulates Hpo signaling in Drosophila (Ribeiro et al, 2010). AP-MS also revealed the regulatory interaction between the FERM domain protein Expanded (Ex) and the co-activator Yorkie (Yki) in Drosophila (Badouel et al, 2009) and the presence of cell-cell junction proteins (AMOT, AMOTL1, AMOTL2) in human YAP and TAZ complexes (Varelas et al, 2010; Wang et al, 2011; Zhao et al, 2011). The few isolated AP-MS studies performed so far used different technologies in different cellular systems, which make it difficult to integrate the available data towards a coherent model of the protein interaction landscape underlying Hpo signaling. Such integrative models are however important, as the control of tissue and organ size can hardly be attributed to just single signaling components, but more likely emerges from concerted molecular events of a complex signaling network. Significant advances in protein complex purification and mass spectrometry instrumentation permit robust characterization of larger groups of complexes and entire pathways, even from human cells (Sardiu et al, 2008; Sowa et al, 2009; Behrends et al, 2010; Glatter et al, 2011; Varjosalo et al, 2013).

Here, we describe the systematic characterization of the protein landscape for the human Hpo pathway. Stringent scoring and cluster analysis of obtained AP-MS data revealed 480 high-confidence interactions among 270 network components that confirm many previously known protein interactions found in humans and interactions of orthologs in other species and provide novel biochemical context for Hpo pathway components. Hierarchical clustering of the obtained interaction data revealed a system of three major signaling modules linked to the transcriptional co-activator YAP1. Aside from the Hpo core kinase complex, the remaining two modules provide multiple links to apico-basal cell polarity (ABCP) and planar cell polarity (PCP). We identified the PP1-ASPP2 module as a regulatory element in controlling transcriptional output of the Hpo pathway and show that polarity proteins differentially bind YAP1 depending on cell-cell contacts. The presented data represents a rich biochemical framework providing 343 previously unidentified high-confidence protein interactions for established pathway components, which will be important for directing future functional experiments to better model tissue growth by the Hpo pathway. The results furthermore support the notion that transcriptional outputs of the major Hpo effector YAP1 may involve a number of biochemical processes linked to cell polarity that may act in parallel to or independent of the canonical Hpo core kinase cassette.

# Results

### Characterization of the human Hpo interaction proteome by a systematic AP-MS approach

To resolve the interaction proteome underlying human Hpo signaling, we selected nine conserved core pathway components as initial baits

for AP-MS analysis in HEK293 cells, based on previously described workflow (Glatter et al, 2009). This initial bait set included the human Hpo kinases MST1 and MST2, the MOB kinase activator 1B (MOB1B), the transcriptional co-activator YAP1, the transcription factor TEAD3, as well as human orthologs of Drosophila Hpo pathway regulators FRMD6 (Willin; homolog of Expanded, Ex), the tumor suppressor protein MERL (homolog of Drosophila Merlin) (Hamaratoglu et al, 2006), and the WW-domain protein KIBRA (Yu et al, 2010). From the interacting proteins identified in this first round of AP-MS experiments we subsequently selected 26 additional secondary bait proteins. Altogether we analyzed 90 AP-MS samples covering 34 bait proteins with at least two biological replicates (Fig 1A and B; Supplementary Table S1). We identified a total of 835 proteins at a protein false discovery rate (FDR) of <1%. To differentiate between high-confidence interacting proteins (HCIPs) and nonspecific contaminants, we filtered our dataset based on $WD^N$-score calculations (Behrends et al, 2010) and relative protein abundance compared to control purification experiments, estimated by normalized spectral counting (Paoletti et al, 2006; Zybailov et al, 2006) (Fig 1C). The data filtering yielded a final network of 270 HCIPs and 480 corresponding interactions (Supplementary Table S2). 88% of these high-confidence interactions have been repeated in all replicate experiments performed (which includes duplicates, triplicates and quadruplicates). Eighty-six percent of the interactions tested by co-immunoprecipitation and Western blotting could be experimentally validated (Supplementary Figure S1), which corresponds well to the experimental validation rate reported in previous large scale AP-MS studies (Sowa et al, 2009; Behrends et al, 2010; Varjosalo et al, 2013). On average, we identified 14.7 HCIPs for each bait, which corresponds to the number of interactors typically found in similar AP-MS studies (Sowa et al, 2009; Behrends et al, 2010; Varjosalo et al, 2013). We also compared the obtained high-confidence AP-MS data set with protein interaction (PPI) data annotated in public databases. 71.5% of our interactions have not been reported at the time of submission and, as expected, the fraction of newly identified PPI varied substantially for the different bait proteins tested (Fig 1E). Well-studied proteins (e.g. MST1, MST2, STRN, STRN3, AMOT, PP1G, RASF1) had a larger fraction of previously annotated interaction partners than less intensively studied proteins (e.g. E41L3, RASF10, RASF9, FRMD5).

Inspection of public PPI data (including yeast two-hybrid and in vitro binding assays) for the 34 baits analyzed in our study resulted in a network of 516 proteins and 719 protein interactions (Supplementary Table S3). 16% of these interactions were found in our AP-MS dataset, which corresponds to 137 known protein interactions. 84.6% of public interactions are reported by a single publication (Supplementary Figure S2A) and the FDR of public PPI data is largely unknown. Therefore we used the number of independent literature reports that support a given interaction as a proxy for data confidence. When we compared our data with a high-confidence subset of public PPI data (>1 publication per interaction) our recall rate increased to 36% (Fig 1D). The fraction of high-confidence public interactions matching with our AP-MS data set was three times higher than the one for the public PPI data not identified in our study, demonstrating the overall robustness of the presented PPI data (Supplementary Figure S2B). Further inspection of the experimental sources of matching public PPI data revealed that two-thirds of the data were obtained by other AP-MS studies (Supplementary Figure S2C). At least 28 independent publications were

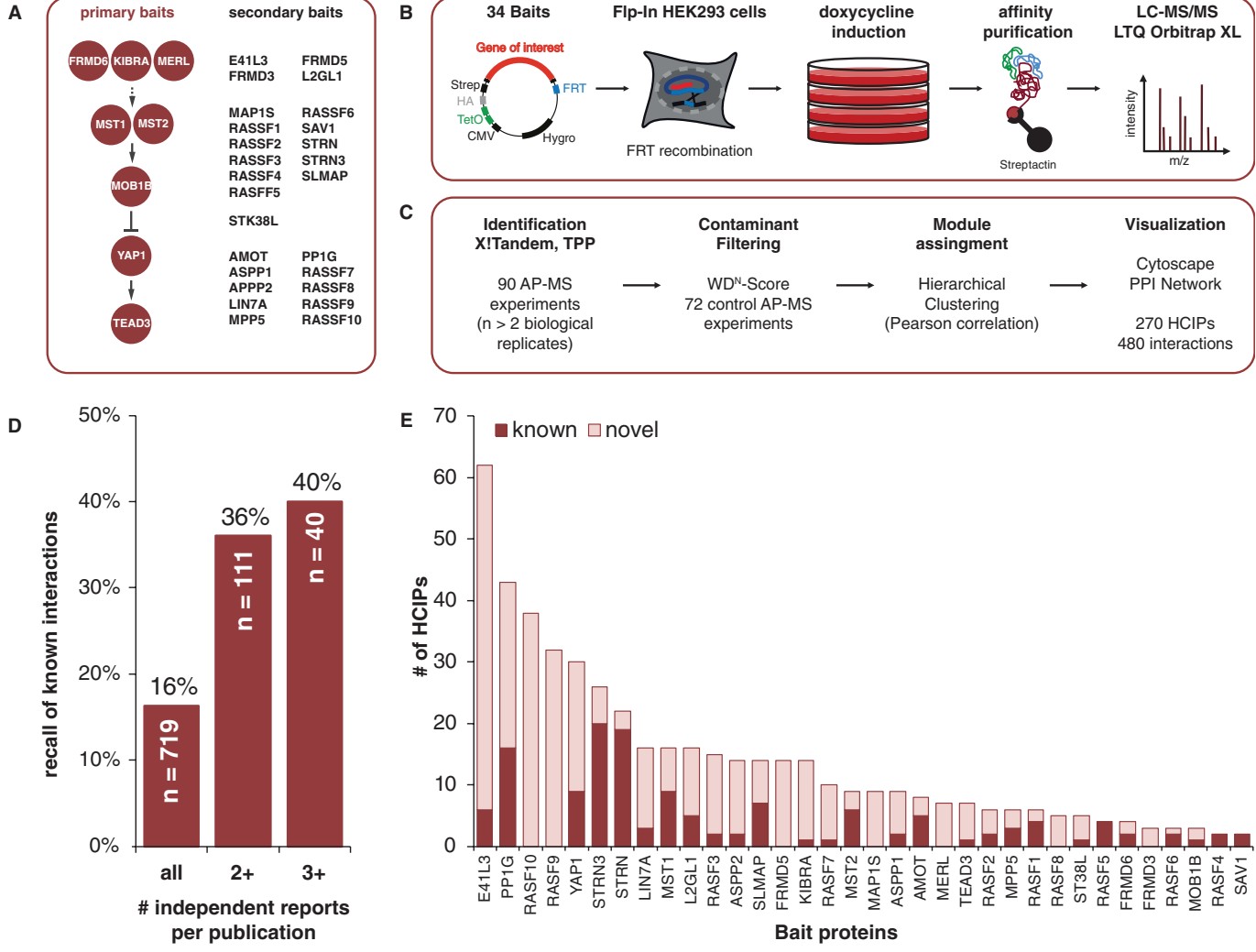

**Figure 1. A systematic affinity purification mass spectrometry (AP-MS) approach to define the human Hpo pathway interaction proteome.**

A   Selection of primary and secondary baits in this study. Baits were selected sequentially, starting with the core components of the Hpo kinase signaling pathway and extended based on obtained AP-MS results or homology to *Drosophila* Hpo components.

B   Biochemical workflow for native protein complex purification from HEK293-Flp T-rex cells. Bait proteins were expressed from a tetracycline-inducible CMV promoter, with a N-terminal Strep-HA fusion tag following induction with doxycycline for 24 h. Cells were lysed, complexes affinity-purified and processed for analysis by tandem mass spectrometry.

C   Data analysis pipeline. Acquired mass spectra from 90 experiments (at least 2 biological replicates per bait) were searched with X!Tandem. Search results were statistically validated by the Trans-Proteomic Pipeline (TPP) to match a protein identification false discovery rate of < 1%. High-confidence interactions were obtained by filtering unspecific binding proteins based on a $WD^N$-score threshold and comparison to control purification experiments (see also Materials and Methods). The resulting high-confidence interactions were hierarchically clustered and visualized.

D   Recall of known interactions from public protein interaction databases. The recall rate was higher, when compared to a more robust subset of literature interactions that required more than one independent report per interaction.

E   Overview on known and novel protein interactions identified in this study. Overall 71.5% of identified interactions have not yet been reported in public databases and each bait associated on average with 14.7 proteins.

needed to cover the annotated 137 interactions also identified in our study. Remarkably, given the collective efforts in the biochemical analysis of this pathway in the past we identified 170 interacting proteins (Supplementary Figure S2D) and 343 interactions for Hpo pathway components that so far were not annotated in public data-bases, which provide important new clues for understanding the molecular mechanisms underlying Hpo signaling in human cells.

## Hierarchical clustering assigns Hpo pathway components to interaction modules

Clustering of bait and prey proteins has been used successfully in the past to infer modular proteome organization from systematic AP-MS data (Sardiu *et al*, 2008). Hierarchical cluster analysis based on pro-tein abundance of prey proteins relative to the corresponding bait protein revealed three major clusters (hereafter referred to as 'mod-

ules'; Fig 2). The first module ('Core Kinase Complex') consists of three interlinked sub-clusters: STRIPAK, a group of complexes containing protein phosphatase 2A linked to Hpo kinase; SARAH, a cluster containing all human SARAH domain proteins, including the human Hpo kinases MST1 and MST2, and finally the MOB1-LATS cluster. Collectively, this module consists of 61 proteins and 143 interactions. The second module ('PP1-ASPP') corresponds to a single large cluster that is enriched for regulatory and catalytic subunits of protein phosphatase 1 (PP1) and contains proteins involved in apico-basal and planar cell polarity (ABCP and PCP). This cluster consists of 90 proteins and 156 interactions. The third module represents a highly interlinked network that consists of a set of Hpo pathway members (L2GL1, DLG1, MERL, KIBRA, YAP1, TEAD) and

proteins linked to ABCP and hence referred to as 'Polarity network' (Fig 2B). This module consists of 72 proteins and 117 interactions.

We next analyzed the occurrence of structural domains (Inter-Pro) across the 270 proteins found in the three modules and performed a hierarchical clustering of these domains across all purifications. The obtained domain cluster mirror the modular organization described above and thus revealed a characteristic enrichment profile of structural domains for each of the modules. Overall we noted a characteristic overrepresentation of specific domains, such as WW, PDZ, FERM, L27 and SARAH (Supplementary Figure S3A and B). Besides the apparent biochemical differences, all three modules converge on the transcriptional co-activator and Yki homolog YAP1, the only protein connecting all three pathway modules.

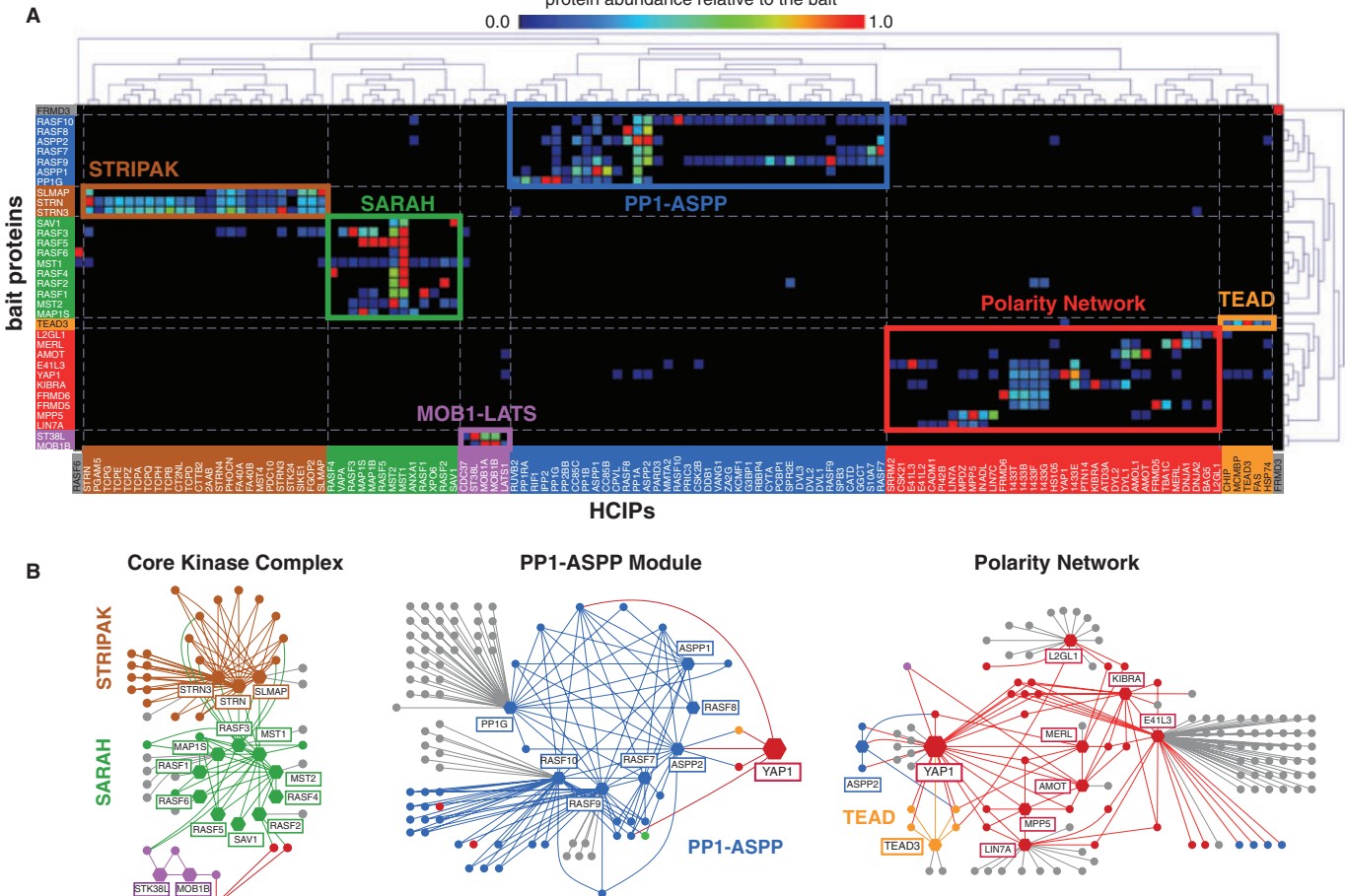

**Figure 2. Hierarchical clustering of bait and HCIPs reveals modular organization of the human Hpo pathway interactome.**

A    Cluster analysis AP-MS data. High-confidence interacting proteins (HCIPs) were clustered based on their abundance relative to the respective bait proteins by an uncentered Pearson correlation algorithm using an average distance metric. Protein abundance relative to the bait is represented by the color-coded squares and corresponds to the legend above the clustergram. Clustering suggests six modules for the Hpo interaction proteome that are organized in three major clusters. Distinct modules are illustrated by colored frames and letters. Color of bait proteins and HCIP's correspond to the respective modules they are part of. The 'core kinase complex' contains the STRIPAK module, the SARAH module containing all SARAH domain proteins, including the Hpo kinase MST1 and MST2, and the downstream kinase LATS1 and its adaptor proteins MOB1A/B ('MOB1-LATS' module). The 'PP1-ASPP module' contains all three PP1 catalytic and several regulatory subunits. The 'polarity network' is defined by multiple polarity complexes, but also contains the transcription factor TEAD3 and the majority of proteins associated with YAP1.

B    Network representation of the obtained modules. The different modules and proteins therein are illustrated by the color scheme used above. Yap1 is the only protein that is common to all three modules.

## Topology of the Hippo core kinase complex

Orthologs of the Drosophila proteins Hpo, Sav and Wts (MST1/2, SAV1, LATS1/2) encode the highly conserved core kinase cassette of the Hpo pathway. Our analysis of the human orthologs of Hpo, Sav and Wts uncovered three major sub-clusters linked to the core kinase module: MOB1-LATS, SARAH and STRIPAK (Fig 3A). MOB1-LATS represents the smallest cluster within this module. It contains the Mats homologs MOB1A and MOB1B which we found in complexes with the protein kinases ST38L and LATS1. LATS1 has been shown to act as a substrate for MST1 but serves also as upstream kinase for the phosphorylation and inactivation of YAP1. Under the experimental conditions applied we identified stable kinase substrate complexes between LATS1 and YAP1, but not between MST1 and LATS1. In MST1/2 complexes we found all human SARAH

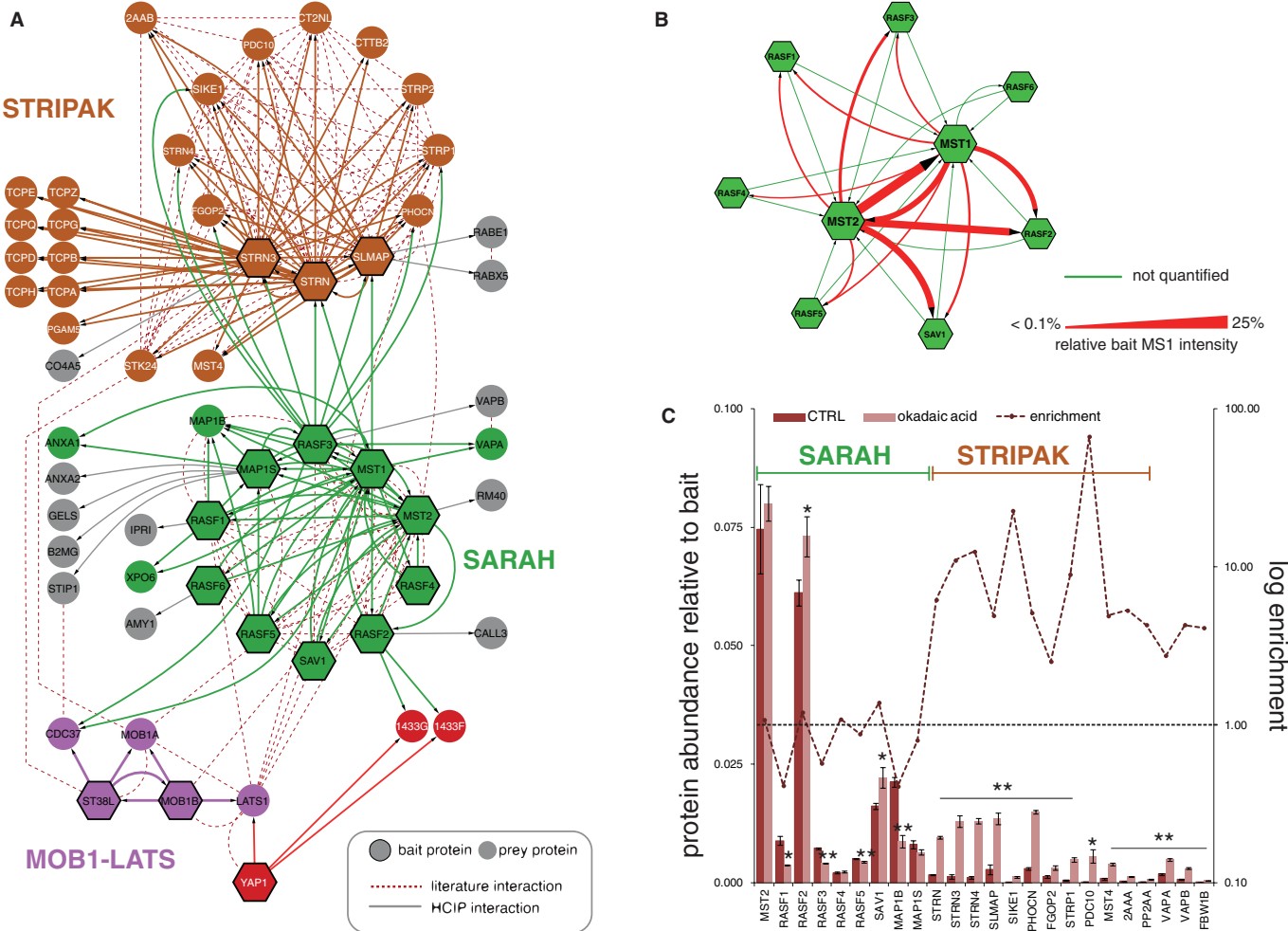

**Figure 3. The human STRIPAK complex associates with RASF3 and MST1/2.**

A  High-resolution interaction map of the human Hpo core kinase cassette and the STRIPAK complex. Bait proteins are indicated as hexagons, prey proteins as circles. Node color corresponds to the modules defined in Fig 2. Solid lines are interactions found by AP-MS in this study, dotted, red lines are obtained from public protein interaction databases. The Hpo kinase homologs MST1 and MST2 interact with SAV1 and all RASF proteins in the SARAH module. The assembly of MOB1A/B and LATS1 forms the downstream kinase cascade of Hpo and associates with the LATS1 substrate YAP1. MST1 and RASF3 interact with the STRIPAK complexes.

B  Interactions between SARAH domain proteins MST1/2 and RASF1–6. All RASF proteins interact with MSTs but not with other RASF proteins or SAV1. MST1/2 form hetero-dimers with each other, as well as the remaining SARAH domain proteins. Interactions identified with MST1/2 were quantified (red lines) by the average intensity of the three most intense precursors ions per protein. The line width represents protein abundance relative to the respective bait. The strongest interactions occur between the MST1/2 heterodimer, whereas the predominant RASF-MST interaction was RASF2 and MST1, or RASF2 and MST2, respectively. Green edges represent interactions that have not been quantified.

C  Abundance changes of interacting proteins of MST1 upon okadaic acid stimulation. HEK293 cells expressing Strep-HA tagged MST1 were treated with 100 nM okadaic acid (OA) for 2 h. Left axis represents the protein abundance relative to MST1. Right axis (log fold change; dotted line) is the logarithmic fold change of the relative abundance of proteins bound to MST1, following OA treatment. The purified MST1 complexes contained an increased amount of STRIPAK associated proteins, whereas SARAH module components only show marginal changes. Similar results were obtained for MST2 (Supplementary Figure S4C). Error bars indicate standard deviation from biological triplicates. Asterisks indicate *t*-test statistical significance (*$P < 0.05$; **$P < 0.01$).

domain-containing proteins which include, apart from MST1/2, the Ras-association domain proteins RASF1-6 and the WW domain protein SAV1 (Fig 3A). It has been proposed that SARAH domain proteins undergo complex formation via homotypic dimerization with other SARAH domain proteins (Scheel & Hofmann, 2003) and some of these interactions have been linked to the regulation of MST1/2 kinase (Praskova *et al*, 2004; Polesello *et al*, 2006; Avruch *et al*, 2009). It is not clear, however, whether the full range of combinatorial possibilities indeed occurs or whether only a subset of dimeric SARAH domain pairs can be formed in human cells. To address this question and to resolve potential differences in concurrent SARAH domain complexes, we included all human SARAH proteins as baits in our AP-MS analysis. In contrast to published protein interaction data (Donninger *et al*, 2011), we observed a high degree of selectivity in binding between the two Hpo kinases and the remaining SARAH domain proteins. Only MST1/2 could undergo combinatorial complex formation with the other SARAH domain proteins, whereas RASF1-6 and SAV1 complexes exclusively contained either MST1 or MST2, but none of the other SARAH proteins (Fig 3B). Based on the average of the three most abundant precursor ion intensities per protein (TOP3) (Silva *et al*, 2006; Rinner *et al*, 2007; Glatter *et al*, 2011) we could quantified the relative abundance of endogenous MST1/2 interactors. Since SH-tagged MST1 and MST2 are expressed to similar levels, which correspond well to the levels of endogenous proteins (Supplementary Figure S4A and B) (Glatter *et al*, 2009) the obtained quantitative data are likely to reflect relative abundances of endogenous MST1 and MST2 complexes. The MST1-MST2 hetero-dimer represents the most abundant SARAH domain assembly, followed by complexes between MST1/MST2 and RASF2 or SAV1 (Fig 3B). This result suggests a system of 15 potentially concurrent SARAH protein sub-complexes and raises the question whether these sub-complexes may have specific biochemical functions. The identification of cellular proteins that bind both RASF family members and MST1/2 may provide a first hint at potential functional diversification of found RASF-MST assemblies. RASF1/3/5, for example, were identified in complexes with MAP1S and MAP1B, two-microtubule-associated proteins also found in MST1/2 complexes, suggesting a potential role for this subset of RASF-MST assemblies in controlling microtubules. In this regard it has been reported that RASF1 stabilizes microtubules through interaction with MAP1 proteins (Dallol *et al*, 2004; Song *et al*, 2005). In contrast, the exportin protein XPO6 co-purified exclusively with RASF1 and MST1, and the vesicle-associated membrane protein-associated protein VAPA and VAPB were found only in RASF3 and MST1 complexes, providing further evidence for potential functional diversification of RASF-Hpo kinase complexes.

In this context the analysis of RASF3 complexes revealed an association with eight members of the human STRIPAK complex. Two of these components (STRN3, SLMAP) were also detected in MST1 complexes, suggesting that human STRIPAK can bind to both MST1, RASF3 or complexes thereof. No STRIPAK components were detected with other RASF family members (Fig 3A). It appears that the STRIPAK-Hpo interaction is conserved since we previously could show that subunits of the *Drosophila* STRIPAK complex also bind to Hpo and dRASSF, where they act as negative regulators of Hpo signaling by recruitment of the protein phosphatase PP2A (Ribeiro *et al*, 2010). The phosphatase inhibitor okadaic acid

(OA) has been shown to activate the human Hpo pathway (Taylor *et al*, 1996; O'Neill *et al*, 2004; Guo *et al*, 2011) and thus may change the composition of MST1/2 complexes. Based on the average TOP3 intensities precursor intensities, we measured the relative abundance of MST1/2-associated proteins in the presence or absence of OA (Fig 3C, Supplementary Figure S4C). Whereas interactions with SARAH domain proteins were largely unaffected or mildly decreased upon OA treatment, we found a strong increase of all STRIPAK subunits associated with MST1/2 in OA-treated cells even though overall amounts of STRIPAK components STRN and SLMAP are not affected by OA as determined by Western blotting (Supplementary Figure 4D). This indicates that under exponential growth conditions, the amount of STRIPAK proteins bound to human Hpo is relatively low compared to the amount of associated SARAH proteins, but may significantly increase upon changes in protein phosphorylation as illustrated by OA treatment (Fig 3C).

By including STRN, STRN3 and SLMAP as baits we could detect all known STRIPAK subunits including the GCK-III subfamily of Ste20 protein kinases, STK24 and MST4, and subunits of the protein phosphatase PP2A. Interestingly, we found that only a specific subtype of the STRIPAK complex containing SLMAP, FGOP1 and SIKE1, but not CTTNBP2 and CTTNBP2NL, binds to Hpo kinases and RASF3. In comparison, the related kinases STK23, STK24 and MST4 are able to bind both subtypes of STRIPAK (Goudreault *et al*, 2009). Whether STRIPAK-mediated recruitment of PP2A leads to dephosphorylation and inhibition of the human Hpo kinases like in *Drosophila* remains to be tested. Our data on the topology of the Hpo kinase core complexes suggest that Hpo kinase, rather than being a single kinase unit, represents a highly dynamic system of multiple concurrent kinase complexes, some of which may be regulated by a specific human STRIPAK-PP2A complex.

### The PP1-ASPP module provides links to apico-basal and planar cell polarity

Apart from the canonical Hpo kinase cassette, we identified a module linked to YAP1 which is centered around the serine protein phosphatase 1 (PP1) and contains several proteins associated with the control of apico-basal cell polarity (ABCP) as well as planar cell polarity (PCP). We initially found PP1 together with ASPP2 (Apoptosis-stimulating of p53 protein 2) in YAP1 complexes. Analysis of ASPP2, its paralog ASPP1, and the PP1G catalytic subunit as AP-MS baits resulted in an extended network which includes all three PP1 catalytic subunits (PP1A, PP1B, PP1G) and multiple regulatory subunits (Fig 4A; Supplementary Table S2). Both ASPP1 and ASPP2 can interact with all PP1 catalytic subunits, the coiled-coil proteins CC85B and CC85C and the Ras-association domain family proteins RASF7, RASF8 and RASF9. We subsequently included RASF7/8/9/10 as baits in our AP-MS analysis. Interaction data from these poorly studied proteins further refined the ASPP/RASF/PP1 sub-network. As ASPP1 and ASPP2 do not interact with each other, we suggest the formation of mutually exclusive ASPP/RASF/PP1 complex isoforms. Whereas all four RASF proteins share the association with PP1 and ASPP, the individual RASF members also form paralog-specific complexes. In both RASF9 as well as RASF10 complexes we found the *Drosophila* homologs of the segment polarity protein Dishevelled DVL1/2/3, together VANG1 and PRIC1/3, which have been implicated in the PCP pathway (Gubb *et al*, 1999; Song *et al*, 2010). RASF9/10 also interact with the alpha and beta subunits of casein

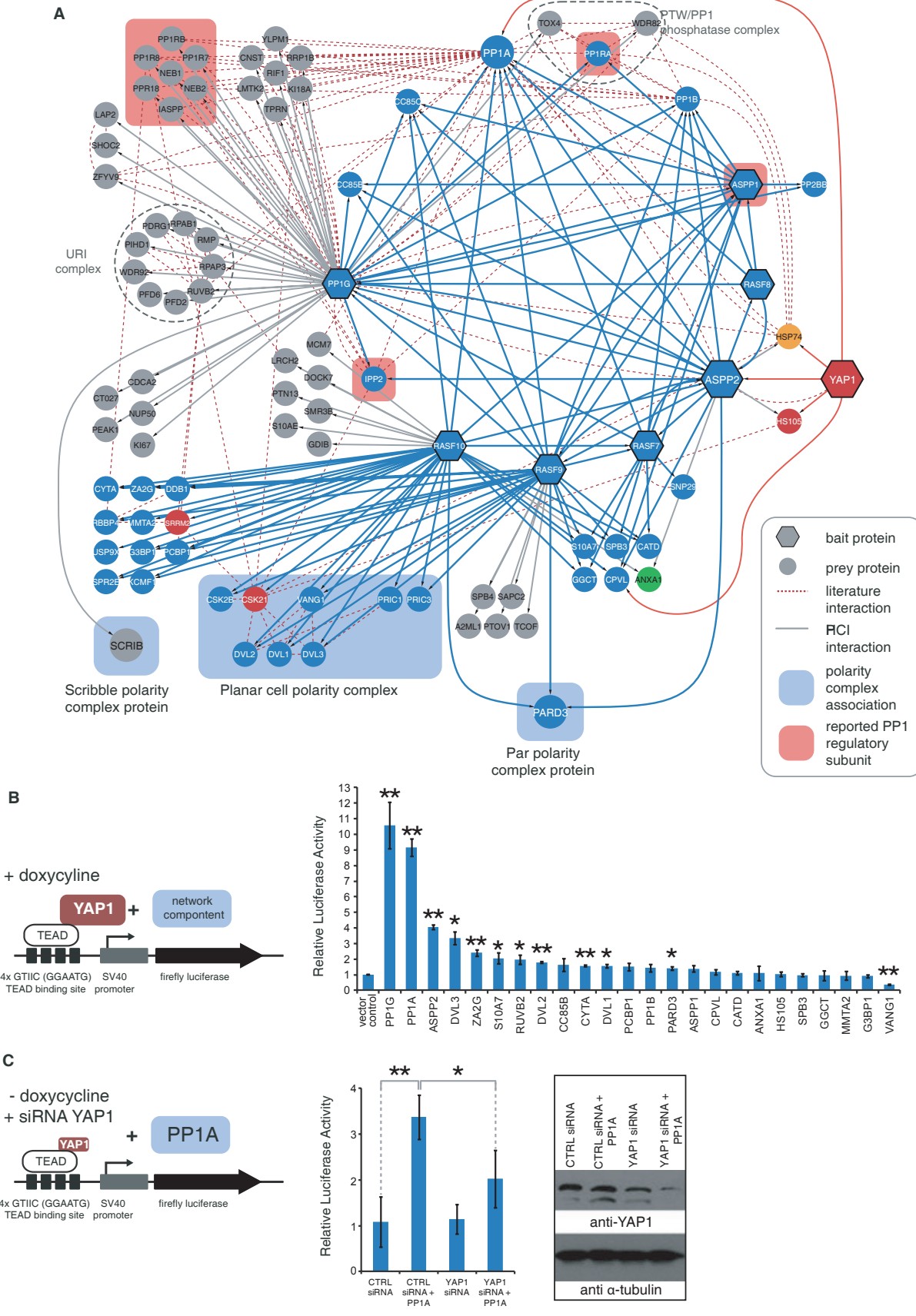

kinase II (CSK21 and CSK2B), the upstream kinases which phosphorylate and activate DVL proteins (Song *et al*, 2000; Bernatik *et al*, 2011).

PARD3, a key apical determinant (St Johnston & Ahringer, 2010), was found in complexes containing ASPP2, RASF9 and RASF10. Such a PP1-PARD3 complex has been reported before, where PP1A directly dephosphorylates PARD3 to stabilize a functional Par/aPKC complex (Traweger *et al*, 2008). PARD3 has been shown to form the Par polarity complex together with PARD6 and the atypical protein kinase C (aPCK), two proteins we identified in L2GL1 complexes described below (Petronczki & Knoblich, 2001). We could not detect any of the RASF9/10-associated cell polarity proteins by using the related RASF7/8 proteins as baits.

Besides the mentioned cell polarity regulators in RASF9/10 complexes, we also found the PDZ protein SCRIB in PP1G phosphatase complexes. SCRIB is the human ortholog of *Drosophila* Scribble, a protein that controls apico-basal polarity together with Lgl and Dlg and has been linked to Hpo signaling (Bilder *et al*, 2000; Cordenonsi *et al*, 2011; Doggett *et al*, 2011). In this regard, Scribble has been genetically linked in *Drosophila* to PP1 phosphatase, via PP1R7 (Sds22) (Jiang *et al*, 2011), a regulatory PP1 subunit we also identified with PP1G. Whether PP1R7 and SCRIB interact together with PP1G in the same sub-complex, remains to be tested.

Our AP-MS data also revealed previously known PP1 sub-complexes including the PTW/PP1 phosphatase complex formed by the PP1G interacting proteins PP1RA, WDR82 and TOX4, which has been shown to regulate chromatin structure during the cell cycle (Lee *et al*, 2010) and the PP1G-associated URI prefoldin complex linked to S6 kinase signaling (Djouder *et al*, 2007). Collectively the presence of the different groups of cell polarity proteins in the PP1-ASPP module suggest a significant role for this module in controlling cell polarity processes by a variety of molecular mechanisms, some of which are likely to involve established Hpo signaling components.

### PP1/ASPP2 complexes promote YAP1 activity

Since we found PP1A and ASPP2 in complexes with YAP1 we wanted to test whether these proteins or other components from the PP1 module might regulate YAP1-dependent transcription. We applied a dual luciferase reporter (DLR) system containing a promoter with TEAD transcription factor binding sites to measure YAP1/TEAD transcriptional activity in response to transient overexpression of PP1 network components in HEK-293-Flp cells

expressing Strep-HA tagged YAP1 (SH-YAP1) (Mahoney *et al*, 2005). We generated a set of constructs for the expression of PP1 network components and validated transgenic expression of 24 components from the PP1-ASPP module by Western blotting (Supplementary Figure S5A). We used MST1 and YAP1 as negative and positive controls to ensure assay specificity (Supplementary Figure S5B). Similar to earlier reports, MST1 decreased (Ota & Sasaki, 2008) whereas YAP1 overexpression increased transcriptional activity of the TEAD reporter (Lamar *et al*, 2012). Upon transient expression of the corresponding transgenes, we found that among the 24 components tested, overexpression of PP1G, PP1A and ASPP2 most strongly enhanced the expression from the TEAD luciferase reporter construct (Fig 4B). The observed activation was dependent on TEAD binding sites in the promoter of the luciferase reporter construct (Supplementary Figure S5C). Remarkably, the related proteins PP1B and ASPP1 had no significant effects. This pattern is in agreement with our interaction data, where we only found ASPP2 and PP1A to bind YAP1 but not ASPP1 and PP1B. We next performed these experiments under conditions where YAP1 expression was silenced by siRNA treatment to test whether the observed activation by PP1 is dependent on YAP1 levels. The results clearly showed that the observed activation of the TEAD luciferase reporter was significantly reduced when YAP1 expression was silenced by siRNA (Fig 4C). In conclusion, the data indicate that PP1A, PP1G, and ASPP2 overexpression increases YAP1-mediated transcriptional activity and suggest a potential role for ASPP2 as the YAP1-interacting determinant of the PP1-ASPP module providing substrate specificity for PP1A and PP1G. Previous work suggests that the ASPP2/PP1 complex can promote TAZ activity by reversing LATS-mediated inhibitory TAZ phosphorylation on Ser-89 and Ser-311 (Liu *et al*, 2011). Our results are consistent with a similar role for ASPP2/PP1 in antagonizing YAP1 inactivation by LATS1/2.

### A cell polarity network linked to L2GL1, KIBRA, MERL and YAP1

The Hpo core kinase cassette has long been recognized as the major upstream regulatory module of YAP1 (Huang *et al*, 2005). Recent studies, however, have indicated multiple modes of Hpo-independent YAP1 regulation, which involve signals from cell-cell contacts (Varelas *et al*, 2010; Wang *et al*, 2011; Zhao *et al*, 2011), mechanical stress (Dupont *et al*, 2011; Wada *et al*, 2011) and cell polarity (Chen *et al*, 2010; Grzeschik *et al*, 2010; Ling *et al*, 2010; Robinson, 2010; Doggett *et al*, 2011), suggesting a role for YAP1 as a major hub for signal integration. Indeed, we found 30

**Figure 4. The ASPP-PP1 module provides links to cell polarity and modulates YAP1 mediated transcriptional activity.**

A   Detailed view of the PP1-ASPP module. The PP1-ASPP network was defined by interaction data from ASPP1/2, PP1G, and RASF7/8/9/10 purification experiments. Three different cell polarity complexes could be linked to the PP1-ASSP network (highlighted in blue). The polarity determinant SCRIB (Scribble) was found with PP1G, the Par polarity complex component PARD3 was identified with RASF9/10 and ASPP2, and proteins linked to planar cell polarity (VANG1, PRIC1/3 and DVL1/2/3) were co-purified with RASF9/10. Node color corresponds to the modules defined in Fig 2.

B   PP1-ASPP network components affect YAP1 transcriptional activity. To test whether overexpression of PP1-ASPP network components could influence TEAD promoter activity, HEK293-Flp cells expressing SH-YAP1 were co-transfected with a firefly luciferase reporter construct containing four TEAD transcription factor binding sites together with constructs for the overexpression of indicated PP1-ASPP network components (left panel). The measured firefly activities were normalized to the activity of a constitutively co-expressing Renilla luciferase and a vector control. Error bars indicate standard deviation from biological triplicates. Asterisks indicate *t*-test statistical significance (\*$P < 0.05$; \*\*$P < 0.01$).

C   PP1A mediated activation of the TEAD promoter activity is dependent on YAP1. Dual luciferase assays have been performed following co-transfection of indicated siRNA and PP1A expression construct. Error bars represent standard deviation from biological triplicates. Asterisks indicate *t*-test statistical significance (\*$P < 0.05$; \*\*$P < 0.01$).

    

HCIPs in YAP1 complexes, 21 of which have not been observed previously. Besides the interactions with canonical Hpo components, including the established upstream inhibitory kinase LATS1 or the transcription factor TEAD3, we identified a large group of proteins linked to cell polarity and cell junction complexes. To gain a more detailed view on the organization of this cell polarity network linked to YAP1 we included several proteins of the cell junction complex (AMOT, MPP5, LIN7A), the FERM domain proteins (FRMD3, FRMD5, FRMD6, E41L3) as baits in our AP-MS experiments. These bait proteins complemented the Hpo components previously linked to cell polarity (L2GL1) or the cell cortex (MERL, KIBRA) (Fig 5).

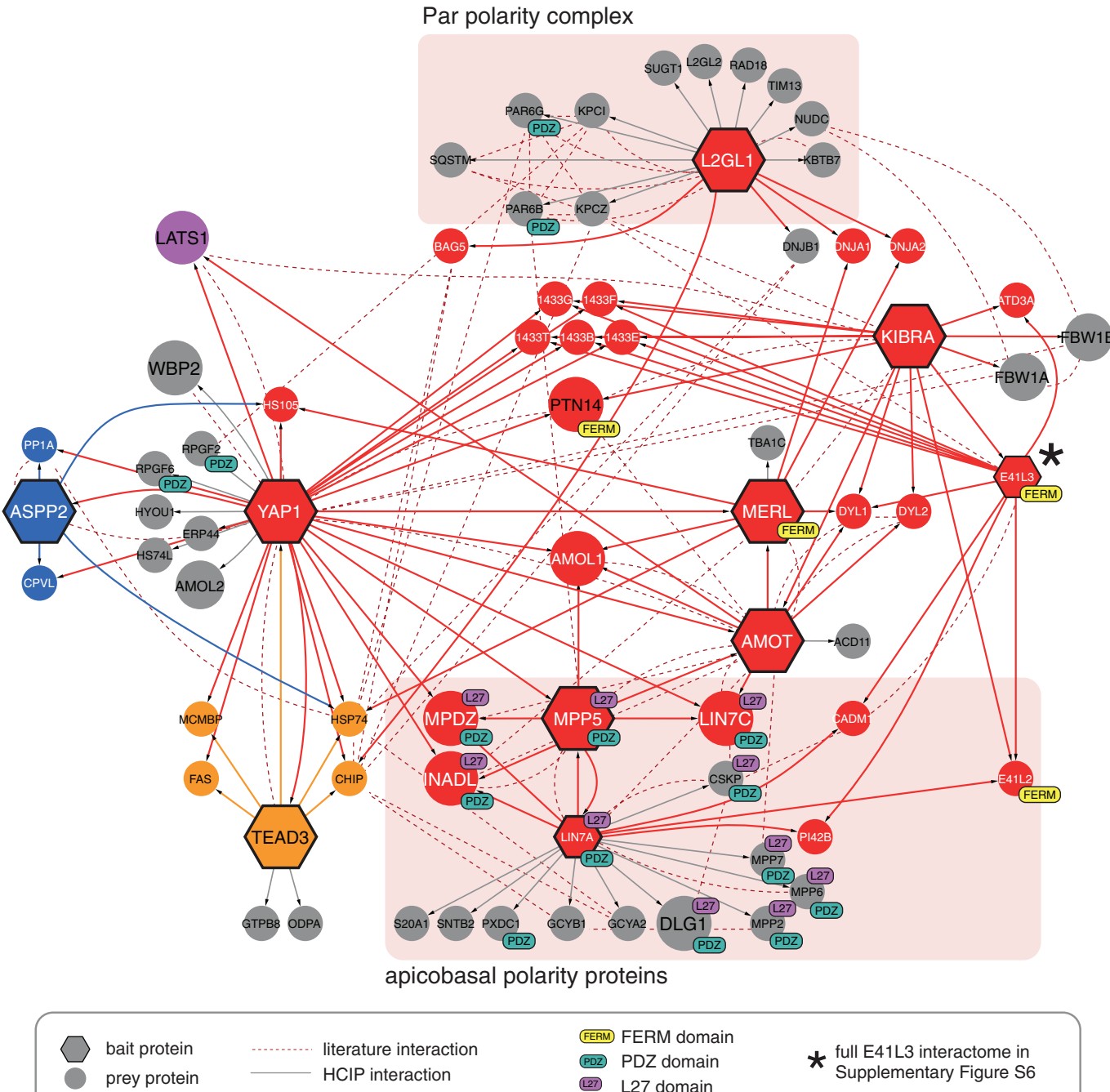

**Figure 5. The YAP1 associated polarity network contains multiple FERM and PDZ domain proteins.**
Besides TEAD3, YAP1 binds to an extended network of FERM, PDZ and L27 domain proteins, which contains AMOT as a central node. AMOT and AMOL1 connect FERM domain proteins with the tight junction associated L27 and PDZ proteins and also directly binds to YAP1. Asterisks (*) refers to supplementary Fig S6. Node and edge color corresponds to the modules defined in Fig 2. Par polarity complexes and apicobasal polarity proteins are indicated with red boxes.

Analysis of complexes containing L2GL1, the human ortholog of *Drosophila* tumor suppressor lethal giant larvae (Lgl), revealed the presence of all major components of the Par polarity complex. They include the aPKC subunits KPCI and KPCZ, SQSTM, a protein previously found associated with aPKC (Sowa *et al*, 2009), and the cell polarity proteins PARD6B and PAR6G. We did not find any interactions between L2GL1 and other canonical Hpo pathway members, however.

FRMD6 is the closest human homolog of the *Drosophila* protein Expanded (Ex). Ex has been shown to associate with Mer and Kibra to from a complex involved in the regulation of Hpo kinase activity (Hamaratoglu *et al*, 2006; Baumgartner *et al*, 2010; Genevet *et al*, 2010; Yu *et al*, 2010). We could not confirm any homologous interactions with FRMD6 (or FRMD3 and FRMD5, Supplementary Table S2) but instead found interactions between YAP1, MERL, KIBRA and the junctional protein AMOT (Angiomotin). YAP1 binds directly through its WW domains to the PPxY motif of Angiomotin and its paralogues (AMOL1 and AMOL2) (Wang *et al*, 2011; Zhao *et al*, 2011). *Drosophila* Ex also has been found to bind the Yki WW domain by its PPxY motif and inhibit Yki activity (Badouel *et al*, 2009; Oh *et al*, 2009). The level of evolutionary Hpo pathway conservation between *Drosophila* and higher eukaryotes is currently of great interest. The similarity in interaction partners has led to the suggestion that AMOT might be the functional homolog of Ex in mammals (Genevet & Tapon, 2011) and a recent evolutionary study concluded that FRMD6 is unlikely to be the functional homolog of Ex, as it lacks a PPxY motif-containing C-terminal domain (Bossuyt *et al*, 2013). These notions are both supported by our experimental data.

Analogous to Ex in *Drosophila*, AMOT forms a highly interlinked network with proteins found in apico-basal polarity complexes (LIN7C, MPP5, MPDZ, INADL), which share a characteristic L27 protein-binding domain (Fig 5). L27 domains form heterotetrameric complexes with each other (Feng *et al*, 2004) and show significant enrichment in our network (Supplementary Figure S2B). When we used LIN7A as bait, we found six additional L27 domain proteins to the already mentioned ones: LIN7A itself and the MAGUK proteins CSKP, MPP2/6/7 and DLG1, the homolog of *Drosophila* Discs large 1 (Dlg). Two other protein binding domains – FERM and PDZ – were also highly overrepresented in our network (Supplementary Figure S3B). The FERM domains are often found in proteins that interface between membrane associated proteins and the cytoskeleton (Chishti *et al*, 1998). Eight of the nine FERM containing proteins from our AP-MS dataset are exclusively found in the cell polarity module. Among the novel PDZ proteins associated with YAP1, we found RAPGEF2/6 that function as GEFs for the Ras GTPase RAP1. RAP1 plays a role in the formation of adherence junctions (AJ) and RAPGEF1 was shown to be required for AJ maturation (Dube *et al*, 2008) which may suggest a link between these processes and the control of YAP1. Other work has suggested interactions between RAPGEF6 and another WW protein, BAG3 which is involved in mechanotransduction (Ulbricht *et al*, 2013).

### Cell-cell contacts control YAP1 complex formation with apico-basal cell polarity proteins

YAP1 nuclear localization is enhanced upon low-density growth or disruption of cell-cell contacts (Zhao *et al*, 2007; Ota & Sasaki, 2008; Varelas *et al*, 2010; Schlegelmilch *et al*, 2011) which raises the questions how YAP1 complex formation may be affected under these conditions. We therefore monitored the abundance of YAP1 interacting proteins in cells subjected to non-adhesive growth over a time course of one hour (Fig 6A), which has been shown to result in YAP1 de-repression (Zhao *et al*, 2012). Based on the average TOP3 intensity (see above) we reliably quantified 20 YAP1-associated proteins across the entire time course. Consistent with the observation that YAP1 nuclear localization is enhanced upon disruption of cell-cell contacts and cell-matrix attachment, we found its association with TEAD3 transcription factor increased when cell are grown in suspension. We also noticed a concerted drop in cell polarity proteins (AMOT, AMOL1, LIN7C, INADL, MPP5 and MPDZ) associated with YAP1 upon disruption of cell-cell contacts. Next we analyzed YAP1 complexes under conditions where cell-cell contacts were reduced by growing cells at low density (Fig 6B). Similar to the results obtained for suspension cell growth these experiments revealed a significant drop of apico-basal cell polarity proteins (AMOT, AMOL1, LIN7C, INADL, MPP5 and MPDZ) when cells were grown at low density. When we analyzed YAP1 dependent activation of the TEAD luciferase promoter we found a gradual increase in activity with decreasing cell density. These results clearly show that the interaction of YAP1 with the cell polarity network is highly dynamic and thus suggests a potential role for the polarity network as an integrated signaling system that controls the transcriptional co-activator YAP1 in response to changes at intercellular junctions.

## Discussion

Metazoan tissue homeostasis at the cellular level emerges from the interplay of concurrent signaling systems that integrate and translate information from neighboring cells, growth factors and mechanical forces to coordinate cell growth, proliferation, apoptosis as well as cell shape changes in a tissue context. Initial models on the regulation of metazoan tissue cell growth by the Hpo pathway were centered on the Hpo core kinase cassette involving a linear array of regulatory relationships among canonical Hpo core components that control the transcriptional effector Yki/YAP. But what are the molecules and mechanisms that link processes relevant for tissue integrity such as mechanotransduction or cell polarity to growth control by the Hpo pathway? Recent genetic and biochemical data on individual pathway components first revealed links between the Hpo pathway and proteins at the cellular membrane and intercellular junction (Piccolo & Cordenonsi, 2013). This suggests close links between epithelial plasticity, cell polarity and growth control by Yki/YAP (Piccolo & Cordenonsi, 2013). However, most of these recent insights were obtained on isolated Hpo pathway components studied in different cellular contexts using different biochemical approaches, which complicates the integration of such information into coherent models of tissue growth.

In this study we present an integrated model on the biochemical landscape underlying human Hpo signaling using an unbiased proteomics approach in a defined cellular context. The integrated view presented here provides several new insights into the global biochemical organization of the Hpo pathway and its relationship to coexisting cell polarity modules. First, human Hpo kinase can be viewed as a system of co-existing kinase sub-complexes, which are mostly based on homotypic SARAH domain interactions involving

 

all nine human SARAH domain proteins. These sub-complexes showed overlapping but also highly specific interactions with other cellular proteins indicating potential functional diversification of

Hpo kinase sub-complexes. Furthermore, human Hpo kinase can bind to STRIPAK, a protein phosphatase 2 complex that binds and negatively regulates Hpo in *Drosophila* (Ribeiro *et al*, 2010). Given

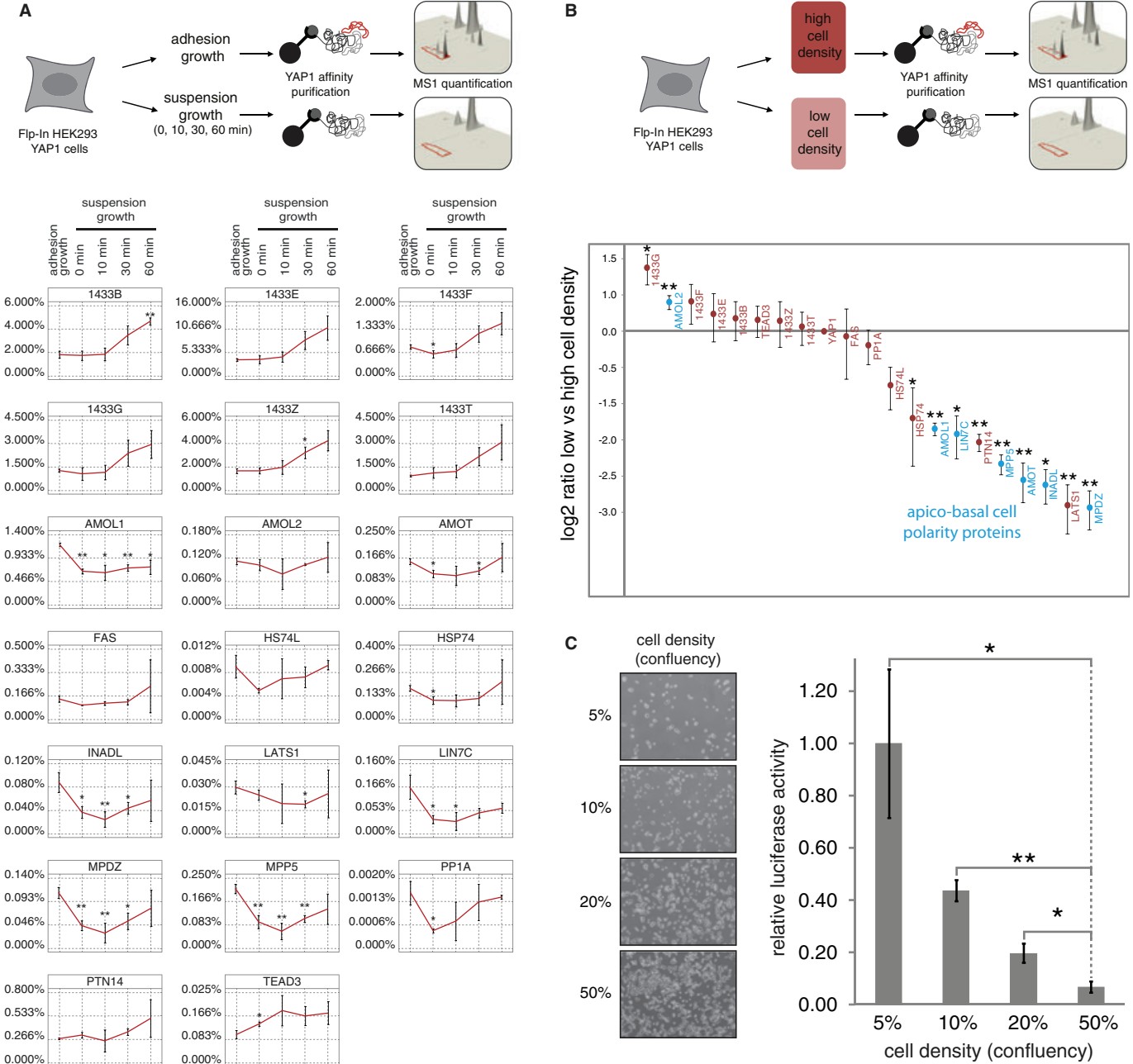

**Figure 6.  YAP1 complex formation is affected by cell-cell contacts.**

A   Schematic overview on the analysis of YAP1 complex dynamics following disruption of cell-cell contacts. SH-YAP1-expressing HEK293 cells are grown on plates or subjected to suspension growth for indicated time points before quantitative AP-MS analysis. Protein abundance relative to YAP1 is measured on the basis of the average intensity of the three most intense peptide precursor ions. The apico-basal polarity proteins AMOT, AMOL1, LIN7C, INADL, MPP5 and MPDZ are decreased, whereas the transcription factor TEAD3 is increased. Average relative abundances of YAP1 associated proteins obtained from three independent experiments are plotted. Error bars represent standard deviation. Asterisks indicate *t*-test statistical significance (*P < 0.05; **P < 0.01).

B   Quantitative AP-MS of YAP1 complexes isolated from cells grown at high and low density. The log2 ratio of measured protein abundances between low and high density conditions was used to visualize YAP1 complex changes. Negative ratios indicate a decrease, positive ratios represent an increase of proteins binding YAP1 at low cell density. Error bars: standard deviation from biological triplicates. Asterisks: *t*-test statistical significance (*P < 0.05; **P < 0.01).

C   Cell density affects TEAD promoter activity in HEK-293 cells. HEK-Flp cells expressing SH-YAP1 were grown at indicated densities and tested for TEAD luciferase promoter activity. Error bars indicate standard deviation from biological triplicates. Asterisks indicate *t*-test statistical significance (*P < 0.05; **P < 0.01).

this evolutionary conservation it remains to be seen whether STRI-PAK may also act as a negative regulator of MST1/2 kinase in human cells. Second, clustering of AP-MS data revealed that cell polarity proteins interacting with Hpo core components can be assigned to two separate modules ('ASPP/PP1' and 'Polarity Network') that are biochemically distinct from the Hpo core kinase module. The modular structure of the Hpo interaction proteome together with our findings that all three modules converge on the major Hpo effector YAP1 is consistent with an emerging view that cell polarity signaling may control YAP1 in parallel or even independent of the canonical Hpo core kinase module. This contrasts models based on genetic studies in *Drosophila* which link cell polarity regulators upstream of Hpo (Hamaratoglu *et al*, 2006; Ling *et al*, 2010). Third, our results also show that the interaction proteome of major Hpo pathway components is highly dynamic and changes in response to an altered cellular environment. In YAP1 complexes we found a strong decrease in abundance of cell polarity proteins upon disruption of cell-cell contacts. These results suggest that cell polarity and cell junction complexes may constitute dynamic signaling modules that relay information on tissue integrity towards the major Hpo effector YAP1.

Besides these general conclusions on the overall organization of the Hpo pathway interaction proteome, this work identified 343 new protein interactions for human Hpo components. These interactions represent an important resource for directing future functional studies to better understand the detailed mechanisms underlying the molecular coupling of cell polarity signaling to cell growth control by the human Hpo pathway.

## Materials and Methods

### Expression constructs

To generate expression vectors for tetracycline-inducible expression of N-terminally Strep-HA-tagged bait proteins, human ORFs provided as pDONR™223 vectors were selected from a Gateway® compatible human orfeome collection (horfeome v5.1, Thermo Fisher Scientific, Waltham, MA, USA) for LR recombination with the destination vector pcDNA5/FRT/TO/SH/GW (Glatter *et al*, 2009). Genes not present in the human orfeome collection were amplified from the MegaMan Human Transcriptome Library (Agilent Technologies, Santa Clara, CA, USA) by PCR, unless stated otherwise (Supplementary Table S1), and cloned into entry vectors by TOPO cloning (pENTR™-TOPO®) or BP clonase reaction (pDONR™223 or pDONR™/Zeo; Life Technologies, Carlsbad, CA, USA).

### Stable cell line generation

HEK Flp-In™ 293 T-Rex cells (Life Technologies) containing a single genomic FRT site and stably expressing the tet repressor were cultured in DMEM medium (4.5 g/l glucose, 2 mM L-glutamine; Life Technologies) supplemented with 10% FCS, 50 µg/ml penicillin, 50 µg/ml streptomycin, 100 µg/ml zeocin and 15 µg/ml blasticidin. The medium was exchanged with DMEM medium (10% FCS, 50 µg/ml penicillin, 50 µg/ml streptomycin) before transfection. For cell line generation, Flp-In HEK293 cells were co-transfected with the corresponding expression plasmids and the pOG44 vector (Life Technologies) for co-expression of the Flp-recombinase using

the FuGENE 6 transfection reagent (Promega, Fitchburg, WI, USA). Two days after transfection, cells were selected in hygromycin-containing medium (100 µg/ml) for 2–3 weeks.

### Protein purification

Stable isogenic cell pools were grown in four 14-cm Nunclon dishes to 80% confluency, induced with 1.3 µg/ml doxycline for 24 h for the expression of SH-tagged bait proteins and harvested with PBS containing 10 mM EDTA. Cells were collected, frozen in liquid nitrogen and stored at −80°C prior to protein complex purification.

The frozen cell pellets were resuspended in 4 ml HNN lysis buffer [50 mM HEPES pH 7.5, 150 mM NaCl, 50 mM NaF, 0.5% Igepal CA-630 (Nonidet P-40 Substitute), 200 µM Na$_3$VO$_4$, 1 mM PMSF, 20 µg/ml Avidin and 1x Protease Inhibitor mix (Sigma-Aldrich, St. Louis, MO, USA)] and incubated on ice for 10 min. Insoluble material was removed by centrifugation. Cleared lysates were loaded on a pre-equilibrated spin column (Bio-Rad, Hercules, CA, USA) containing 50 µl Strep-Tactin sepharose beads (IBA GmbH, Göttingen, Germany). The beads were washed two times with 1 ml HNN lysis buffer and three times with HNN buffer (50 mM HEPES pH 7.5, 150 mM NaCl, 50 mM NaF). Bound proteins were eluted with 600 µl 0.5 mM biotin in HNN buffer. To remove the biotin, eluted samples were TCA precipitated, washed with acetone, air-dried and re-solubilized in 50 µl 8 M urea in 50 mM NH$_4$HCO$_3$ pH 8.8. Cysteine bonds were reduced with 5 mM TCEP for 30 min at 37°C and alkylated in 10 mM iodoacetamide for 30 min at room temperature in the dark. Samples were diluted with NH$_4$HCO$_3$ to 1.5 M urea and digested with 1 µg trypsin (Promega) overnight at 37°C. The peptides were purified using C18 microspin columns (The Nest Group Inc., Southborough, MA, USA) according to the protocol of the manufacturer, resolved in 0.1% formic acid, 1% acetonitrile for mass spectrometry analysis.

### Mass spectrometry

LC-MS/MS analysis was performed on a LTQ Orbitrap XL mass spectrometer (Thermo Fisher Scientific). Peptide separation was carried out by a Proxeon EASY-nLC II liquid chromatography system (Thermo Fisher Scientific) connected to an RP-HPLC column (75 µm x 10 cm) packed with Magic C18 AQ (3 µm) resin (WICOM International, Maienfeld, Switzerland). Solvent A was used as RP-HPLC stationary phase (0.1% formic acid, 2% acetonitrile). Solvent B (mobile phase; 0.1% formic acid, 98% acetonitrile) was used to run a linear gradient from 5 to 35% over 60 min at a flow rate of 300 nl/min. The data acquisition mode was set to obtain one high resolution MS scan in the Orbitrap (60,000 @ 400 m/z). The 6 most abundant ions from the first MS scan were fragmented by collision-induced dissociation (CID) and MS/MS fragment ion spectra were acquired in the linear trap quadrupole (LTQ). Charge state screening was enabled and unassigned or singly charged ions were rejected. The dynamic exclusion window was set to 15 s and limited to 300 entries. Only MS precursors that exceeded a threshold of 150 ion counts were allowed to trigger MS/MS scans. The ion accumulation time was set to 500 ms (MS) and 250 ms (MS/MS) using a target setting of $10^6$ (MS) and $10^4$ (MS/MS) ions. After every replicate set, a peptide reference sample containing 200 fmol of human [Glu1]-Fibrinopeptide B (Sigma-Aldrich) was analyzed to monitor the LC-MS/MS systems performance.

                                                

## Protein identification

Acquired spectra were searched with X!Tandem (Craig & Beavis, 2004) against the canonical human proteome reference dataset (http://www.uniprot.org/), extended with reverse decoy sequences for all entries. The search parameters were set to include only fully tryptic peptides (KR/P) containing up to two missed cleavages. Carbamidomethyl (+57.021465 amu) on Cys was set as static peptide modification. Oxidation (+15.99492 amu) on Met and phosphorylation (+79.966331 amu) on Ser, Thr, Tyr were set as dynamic peptide modifications. The precursor mass tolerance was set to 25 ppm, the fragment mass error tolerance to 0.5 Da. Obtained peptide spectrum matches were statistically evaluated using PeptideProphet and protein inference by ProteinProphet, both part of the Trans Proteomic Pipeline (TPP, v.4.5.1) (Deutsch *et al*, 2010). A minimum protein probability of 0.9 was set to match a false discovery rate (FDR) of <1%. The resulting pep.xml and prot.xml files were used as input for the spectral counting software tool Abacus to calculate spectral counts and NSAF values (Fermin *et al*, 2011).

## Evaluation of high confidence interacting proteins (HCIP)

Adjusted NSAF values of identified co-purified proteins were compared to a mock AP control dataset consisting of 62 StrepHA-GFP or 12 StrepHA-RFP-NLS purification experiments. GFP and RFP control datasets have been deposited in the CRAPome contaminant repository for affinity purification (Mellacheruvu *et al*, 2013). The protein abundance in the control dataset was estimated by averaging the 10 highest NSAF values per protein among all 74 measurements. In order for candidate interactions to pass high confidence filtering, the enrichment threshold over the control dataset was set to >10. Adjusted NSAF values were also used to calculate $WD^N$-scores of all the potential interactions (Behrends *et al*, 2010). A simulated data matrix was used to calculate the WD-score threshold below which 95% of the simulated data falls. All raw WD-scores were normalized to this value. All interactions that have a $WD^N$-score greater or equal to 1 passed this filtering step. From the high confidence interaction dataset (control ratio and $WD^N$-score) a gene distance matrix (GDM) was generated, based on an uncentered Pearson distance metric, using the software tool Multi-Experiment Viewer (Saeed *et al*, 2003) (http://www.tm4.org/mev/). To increase sensitivity, the obtained distances were mapped to the unfiltered PPI dataset and sub-threshold interactions were rescued, if the distance was greater than zero ($n = 44$ protein interactions).

## Label free quantification of MST1 and YAP1 interactions

HEK Flp-In™ 293 T-Rex cells (Life Technologies) expressing SH-MST1 and SH-MST2 were treated with 100 nM okadaic acid (LC Laboratories, Woburn, MA, USA) for 2 h. Affinity purification and mass spectrometry analysis was carried out as described above.

For cell density AP-MS analysis of YAP1 complexes, HEK Flp-In™ 293 T-Rex cells (Life Technologies) expressing SH-YAP1 were either harvested at 30% (low density) or 80% (high density) confluency. For the suspension time course experiments the SH-YAP1 expressing cells were grown to 80% confluency and forced into suspension by treatment with Trypsin-EDTA (Life Technologies) and kept on a tube rotator for 0, 10, 30, and 60 min. The cells were pelleted by centrifugation, washed with PBS and frozen in liquid nitrogen to be stored at

−80°C prior to protein complex purification. The affinity purification and mass spectrometry analysis was performed as described above.

Label-free quantification to estimate relative protein abundances was performed by averaging the three most intense peptide precursor ions for identified proteins, using the commercial software tool Progenesis LC-MS (Nonlinear USA Inc., Durham, NC, USA). Raw abundance values were normalized to the bait protein intensity.

## Network visualization and accessed public protein interaction databases

Protein Interaction data was visualized with Cytoscape 2.8.3 (http://www.cytoscape.org) (Shannon *et al*, 2003). Known interactions were obtained from the protein interaction network analysis platform PINA (http://cbg.garvan.unsw.edu.au/pina/) (Wu *et al*, 2009), for the bait proteins used in this study.

## Dual luciferase reporter assay

Flp-In HEK293 SH-YAP1 cells were grown to 50% confluency in a six-well plate format. Where indicated, YAP1 expression was induced with 500 ng/ml doxycline for 6 h prior transfection. The cells were transfected with 80 ng pGL3-4xGTIIC-49 (firefly luciferase reporter with TEAD binding sites) or pGL3-49 (negative control; no TEAD binding sites), and 0.3 ng pRL-CMV (Renilla luciferase; Promega) using the transfection reagent FuGENE 6 (Promega). For protein overexpression, 100 ng of pDEST40 (Life Technologies) expressing the corresponding V5-tagged Hpo network components were co-tranfected with the luciferase plasmids. The transfected cells were kept under doxycycline induction for the next 24 h. For RNAi treatment the cells were transfected with 50 nM YAP1 or control Silencer® Select siRNA (Life Technologies) 24 h prior transfection of the expression constructs using FuGENE HD (Promega), DMEM and FBS was removed and the cells were washed with PBS. Cell lysis for the Dual-Luciferase Reporter Assay (DLR; Promega) were performed according to the manufacturer's instructions. The luciferase signals were measured with a Synergy HT Multi-Mode microplate reader (BioTek Instruments Inc., Winooski, VT, USA). All obtained values were normalized to the Renilla activity.

For cell density DLR experiments, Flp-In HEK293-SH-YAP1 were induced with 500 ng/ml doxycline for 24 h and harvested at 5, 10, 20 and 50% confluencies. The DLR assay was performed according the manufacturer's instructions. Cell densities were estimated using the software tool ImageJ (http://rsbweb.nih.gov/ij/) (Schneider *et al*, 2012).

## Data deposition

The mass spectrometry data from this publication have been submitted to the PeptideAtlas database (http://www.peptideatlas.org/) and assigned the identifier PASS00281. The protein interactions from this publication have been submitted to the IMEx (Orchard *et al*, 2012) consortium through IntAct (Aranda *et al*, 2010) (http://www.ebi.ac.uk/intact/) and assigned the identifier IM-20985.

**Supplementary information** for this article is available online: http://msb.embopress.org

## Acknowledgements

We would like to thank Y. Zhou, I. Farrance, M. Wehr, J. Kremerskothen and A. Barnekow for providing expression and reporter constructs. This work was sup-

ported by the European Union 7th Framework project, PROSPECTS (Proteomics Specification in Space and Time, grant HEALTH-F4-2008-201648) and ERCETH Zurich internal funds to RA, the European Union 7th Framework project, SYB-ILLA (Systems Biology of T-cell activation) to SH and MG and European Union 7th Framework Marie Curie Actions IEF grant 'Cancer Kinome' (grant no. 236839) to MV.

## Author contributions

SH designed experiments, carried out LC-MS analysis, analyzed the data, prepared the figures and wrote the manuscript, AW, AD and MV contributed to reagents, NT and RA contributed to study design and commented on the manuscript, MG designed the study, analyzed the data and wrote the manuscript.

## Conflict of interest

The authors declare that they have no conflict of interest.

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
