## [Review Process File · Molecular Systems Biology]

Interaction Proteome of Human Hippo Signaling: Modular Control of the Co-activator YAP1

Simon Hauri, Alexander Wepf, Audrey van Drogen, Markku Varjosalo, Nic Tapon, Ruedi Aebersold, Matthias Gstaiger

Corresponding author: Matthias Gstaiger, ETH

Review timeline:

Submission date:	31 July 2013
Editorial Decision:	12 September 2013
Revision received:	13 November 2013
Accepted:	20 November 2013

Editor: Maria Polychronidou

Transaction Report:

1st Editorial Decision

12 September 2013

Thank you again for submitting your work to Molecular Systems Biology. We have now heard back from the three referees who agreed to evaluate your manuscript. As you will see from the reports below, while the reviewers mention that your work provides limited novel insights into the Hippo pathway, they acknowledge that it presents a potentially useful resource. Overall, the reviewers raise a series of concerns on your work, which should be convincingly addressed in a revision of the manuscript.

Without repeating all the points listed below, among the more fundamental issues are the following:

- The predicted interactions need to be validated using independent methods.
- Additional experimentation is required to convincingly demonstrate the involvement of the novel interactors in YAP1-dependent transcriptional regulation. Reviewer #3 provides constructive suggestions in this regard, i.e. assessing the effect of PP1G, PP1A and ASPP2 in absence of YAP1. The ChIP-Seq and in vitro experiments suggested by this reviewer are not mandatory, even though we would welcome inclusion of such data if available.
- Additional validations and controls should be provided for the interactions of YAP1 with components of the cell polarity network.
- Reviewer #1 lists a series of technical concerns that need to be carefully addressed.

If you feel you can satisfactorily deal with these points and those listed by the referees, you may wish to submit a revised version of your manuscript. Please attach a covering letter giving details of the way in which you have handled each of the points raised by the referees. A revised manuscript will be once again subject to review and you probably understand that we can give you no guarantee at this stage that the eventual outcome will be favorable.

Reviewer #1:

Hauri et al. report the use of quantitative AP-MS to characterize the interaction network around nine components of the Hippo pathway. Subsequently, many preys were themselves used as baits. This resulted in a network implying 270 proteins and 480 high confident interactions. A very significant fraction of the identified interactions were novel. Hierarchical clustering revealed that these proteins assembled in three major signalling modules: the core Hpo kinase complexes, the apico-basal cell polarity (ABCP) and the planar cell polarity (PCP). The work complements previous efforts (also from the authors). The originality of the manuscript resides in the fact that the authors have quantified some of the interactions. This led for example, to the identification of series of combinatorial SARAH complexes and the confirmation that the stoichiometries of components of SARAH complexes were affected by OA treatment whereas the ones of the STRIPAK complexes were largely unaffected. The authors used state-of-the-art mass spectrometry; they are expert in AP-MS. The dataset is likely to be useful for the scientific community. Several important points need to be addressed:

- A first point concerns the assessment of data quality. What is the reproducibility? Also the coverage of previous (literature) knowledge concerns apparently only interactions (i.e. only new interactions amongst known members?). Were new components of the Hippo pathway identified? If yes, then the authors should clearly mention what they are, i.e. proteins of unknown function, etc.

Are there known components of the pathways that have been missed?

- Another issue concerns protein abundance and over-expression of the baits. This is obviously a key point as the authors discuss subcomplexes and protein stoichiometries. Were all baits expressed at similar levels, i.e. can we compare different complexes? Were they expressed at levels similar to the endogenous (untagged) version, i.e. does the stoichiometry reflect physiology (or is it an artefact of overexpression)? To what extent do variations in the level of expression of the baits affect the measured complexes stoichiometries? In other words does the graph in Figure 3B (for example) reflect different baits abundances rather than different complex stoichiometries? Similarly does OA affect bait, prey abundances?

- The authors used a method designed to characterize protein complexes, they however frequently reduce the dataset to a set of binary interactions. For example in Figure 3A they label the edges as "NHCHIP interactions" this is formally probably wrong, as AP-MS does not give information on direct (physical) interactions. This may require rethinking (or at least relabeling). Similarly the authors should be very careful in the interpretation of "novel interactions" as the method does not allow the direct charting of physical interactions.

- The literature quoted is biased towards the author's own contribution. For example, they are not the ones who first described the use of AP-MS for the charting of protein complexes. This was done first by Bertrand Seraphin in 1999! It would be great to see a more balanced (and fair) reference list.

Minor points:

- Supplementary Table 3 should also include the source or reference (PubMed ID, etc)

Reviewer #2:

The Hippo pathway plays an important and conserved role in controlling organ growth. This manuscript describes identification by mass spectrometry of a Hippo-pathway interactome, obtained by using several known components as baits. The authors also performed some limited functional validation of the ability of some of the interacting proteins they identified to modulate Hippo signaling when over-expressed in cultured cells, using a transcriptional reporter assay. The characterization of part of the interaction network under different conditions of cell attachment is also a nice addition. The manuscript does not provide compelling new insights into the pathway, but describes a screen that identifies candidate new players and interacting modules. I think the network of physical interactions identified by these studies is an interesting and potentially valuable resource for Hippo pathway research, and hence will be of general interest.

In analyzing the interacting proteins obtained, the authors emphasize the expected interactions they identified along with the novel interactors. I would find it of interest if they could also comment on any known interactors of the baits they used that were not identified in their studies.

Reviewer #3:

This manuscript describes the analysis of the human Hippo growth regulatory pathway. The authors describe the proteomic analysis of this network and they obtain 480 protein protein interactions between 270 network components. The authors use standard and established quantitative proteomic methods to analyze this network. The main biological aspects of the networks that the authors describe in part is the impact of interacting proteins on a reporter assay of the transcriptional activator YAP1 and differences in YAP1 interactions when cells are grown under adhesion or suspension conditions. The data in this manuscript is appropriately generated using standard approaches, and there is some follow up biology.

However, the lack of novel methods or mechanistic/functional insight into the network describe results in this manuscript falling short of what is expected from a manuscript in Molecular Systems Biology. This is particularly evident if one compares this manuscript to the recently published MSB paper on the human HDAC protein interaction network from the Cristea group. In that paper, several other methods were used to validate the network and provide novel insights into the network. This included the use of imaging to determine colocalization of novel interacting partners, co-immunoprecipitation to validate novel interactions, and a clever use of SAINT scores and I-Dirt stability ratios to determine the relative interaction stability of HDAC-containing complexes.

This is the major issue with the current manuscript on the human Hippo pathway. Essentially, it provides a reported high confidence list of interactors, but there is no validation of these interactors with methods other than proteomic methods. Co-localization of components with each other co-immunoprecipitation studies are valuable to provide greater confidence in results to a wider audience of researchers rather than proteomics/systems biology researchers alone. Also, there are no novel methods that are present in the current manuscript that would potentially increase the value of this manuscript.

One area of follow up biology that would need extensive additional work that would raise the interest of this manuscript relate to luciferase based assay regarding YAP1 activity stimulation. This data is in Figure 4B. Here the authors show that in this assay, PP1G, PP1A, and ASPP2 stimulate luciferase activity. There are a couple of issues with the way this experiment is described that first need to be clarified. Based on the figure legend and the manuscript it is unclear if these proteins are stimulating activity or maintaining activity. The addition of YAP1 to the assay has the highest level of activity followed by PP1G, PP1A, and ASPP2. Are the authors assuming that in this assay endogenous YAP1 is present? Is YAP1 the only transcription factory that binds TEAD sites? Having a completely in vitro way to test the effect of these proteins on YAP1 activity would benefit these results since it is not clear that these proteins are stimulating activity via YAP1. Perhaps another control to add to this would be knocking down YAP1 in these cells and then seeing if PP1G, PP1A, and ASPP2 still stimulate activity. In addition, do these three proteins localize with YAP1 in cells? Here imaging studies would be valuable to co-localize these proteins in the nucleus, for example. Finally, to provide additional data to support this potential mechanism, ChIP-Seq should be done with YAP1, PP1G, PP1A, and ASPP2 to see if they co-localize DNA in the cell. These datasets would greatly strengthen this manuscript

The other area of biology that could be expanded greatly is the role of YAP1 interactions under adhesion and suspension growth. However, currently this data is problematic. The authors need to provide statistical significance for these results and provide the justification for this. Certainly the decrease in interaction of AMOT, AMOL1, and AMOL2 are dramatic. Following up on these results would also strengthen the manuscript. Here again, imaging and PCR would potentially be valuable. Does the localization of YAP1 and these three proteins change in the cells or is YAP1 somehow affecting the transcription levels of these three proteins? Here again, more mechanistic and functional information would be highly valuable to strengthen this manuscript.

Reviewer #1:

Hauri et al. report the use of quantitative AP-MS to characterize the interaction network around nine components of the Hippo pathway. Subsequently, many preys were themselves used as baits. This resulted in a network implying 270 proteins and 480 high confident interactions. A very significant fraction of the identified interactions were novel. Hierarchical clustering revealed that these proteins assembled in three major signalling modules: the core Hpo kinase complexes, the apico-basal cell polarity (ABCP) and the planar cell polarity (PCP). The work complements previous efforts (also from the authors). The originality of the manuscript resides in the fact that the authors have quantified some of the interactions. This led for example, to the identification of series of combinatorial SARA complexes and the confirmation that the stoichiometries of components of SARA complexes were affected by OA treatment whereas the ones of the STRIPAK complexes were largely unaffected. The authors used state-of-the-art mass spectrometry; they are expert in AP-MS. The dataset is likely to be useful for the scientific community. Several important points need to be addressed:

- *A first point concerns the assessment of data quality. What is the reproducibility?*

88% of the high confidence interactions listed in the presented Hippo study have been repeated in all replicate experiments performed (which includes duplicates, triplicates and quadruplicates). Details on the number of replicate experiments and number of repeat identifications are summarized in Supplementary Table S2. These numbers correspond well to the recently reported intra-lab reproducibility for the applied AP-MS workflow and to the results obtained in earlier studies where we described the method for the first time (Varjosalo et al. 2013, Nature Methods, Glatter et al. 2009, MSB) and are referenced at page six of the revised manuscript.

Also the coverage of previous (literature) knowledge concerns apparently only interactions (i.e. only new interactions amongst known members?). Were new components of the Hippo pathway identified? If yes, then the authors should clearly mention what they are, i.e. proteins of unknown function, etc. Are there known components of the pathways that have been missed?

The components of the “Hippo pathway” are not precisely defined in the literature which makes it difficult to judge what is an accepted Hippo pathway member and what not. To circumvent this problem we have compared all identified binding partner for those Hpo components used as baits in our study and compared them with already known binding partners of the same Hpo signaling proteins. This has been addressed in Figure 1 on a more general level where we describe recall and overlap with existing protein interaction information. We have compared our data with literature information in the revised Supplementary Figure S2 and included a short description for each protein in the Supplementary Table S2. In addition we modified Supplementary Table S3, which lists now all previously annotated interactors for the bait proteins used in our study and indicates which interactors we did not identify. Overall we identified 170 novel interacting proteins and 409 proteins reported in the public databases were not identified (Supplementary Figure S2D). These changes are referenced in the revised text on page seven.

- *Another issue concerns protein abundance and over-expression of the baits. This is obviously a key point as the authors discuss subcomplexes and protein stoichiometries. Were all baits expressed at similar levels, i.e. can we compare different complexes?*
- *Were they expressed at levels similar to the endogenous (untagged) version, i.e. does the stoichiometry reflect physiology (or is it an artefact of overexpression)?*

We thank the reviewer for pointing this out. We measured changes in relative protein abundance using peptide precursor ion intensities in our study but do not claim precise stoichiometries given the known limited accuracy in inferring absolute protein amounts from peptide ion intensities. In Figure 3B we estimated the relative abundance of MST1/2 containing complexes in human cells by combining network topology information of MST1/MST2 associated proteins with quantitative interaction data using MST1 and MST2 as bait proteins. To exclude potential influence of bait protein expression levels on our quantitative results we followed the reviewers suggestions and

measured tagged MST1 and MST2 protein levels by Western blotting using anti HA antibodies. The results showed that the levels of tagged MST1 and MST2 used for AP-MS are very similar (see Supplementary Figure S4A). We previously showed that the expression of tagged proteins including MST2 in the HEK293 cell lines used for AP-MS is moderate and corresponds well to endogenous protein levels (Glatter et al., MSB, 2009, Figure 1D). We repeated these experiments to include also MST1 and confirmed that tagged MST1 levels are similar to the corresponding endogenous proteins level as well. The results from these experiments are referenced in the text on page nine and are now included in Supplementary Figure S4A.

- *To what extent do variations in the level of expression of the baits affect the measured complexes stoichiometries?*

We want to emphasize again that in our study on the Hpo network we do not claim precise stoichiometries but changes in relative protein abundance. We have addressed this point however in a previous study using isotopically labeled reference peptides for absolute protein abundance measurements (Wepf et al., Nature Methods 2009, Supplementary Figure 3). We could show for a number of different bait proteins that the measured stoichiometries are robust over a range of bait expression levels using the same HEK293 expression system that was used also in the Hippo study.

- *In other words does the graph in Figure 3B (for example) reflect different baits abundances rather than different complex stoichiometries?*

The quantitative information in the graph of Figure 3B represents relative abundances of endogenous proteins in complexes with the bait proteins MST1 or MST2 (red edges). As shown above the protein level of exogenous MST1 and MST2 used for quantitative AP-MS are comparable and similar to the corresponding endogenous proteins and thus the estimated subcomplex abundances are likely to reflect the physiological situation. This is also reflected by our finding that the order in relative abundance of proteins associated with the highly homologous Hippo kinases MST1 and MST2 is similar as well (RAS2>SAV1>MAP1B>RAS3). The green edges in Figure 3B do not contain quantitative information and were derived from AP-MS using SAV1 and the RAS1, RAS2, RAS3, RAS4, RAS5 and RAS6 as bait proteins to provide mutual exclusive interaction information for inferring MST1/MST2 subcomplex models. We have better explained this in the revised Figure legends to Figure 3B.

Similarly does OA affect bait, prey abundances?

We have analyzed the protein levels for the bait protein MST1 as well as for prey protein STRN and SLMAP upon OA treatment using Western blotting. Although we found a marked increase in STRN and SLMAP in MST1 complexes following OA treatment we did not observe a change in the overall expression levels of these proteins following OA treatment. Therefore we believe that at least the observed increased binding of the STRIPAK component STRN and SLMAP to MST1 cannot be explained by changes in protein abundance and thus may involve other mechanisms such as protein phosphorylation. It is however clear that the limited availability of high quality antibodies for the proteins identified by mass spectrometry does not allow a general conclusion on the mechanisms underlying the OA mediated changes in human Hippo complexes. Results from these experiments are shown in Supplementary Figure S4D and are mentioned in the revised manuscript on page 10.

Second

- *The authors used a method designed to characterize protein complexes, they however frequently reduce the dataset to a set of binary interactions. For example in Figure 3A they label the edges as „HCIP interactions" this is formally probably wrong, as AP-MS does not give information on direct (physical) interactions. This may require rethinking (or at least relabeling). Similarly the authors should be very careful in the interpretation of "novel interactions" as the method does not allow the direct charting of physical interactions.*

In the text we do not claim any direct interactions since this is not possible from AP-MS data as correctly pointed out by reviewer 1. This point has been addressed repeatedly in AP-MS studies.

Like most previous systematic AP-MS studies we use the spokes model to represent our interaction data, which does not differentiate between direct or indirect interactions in our network graphs respectively in our comparison to the published interaction information. Also the interaction information annotated from previous AP-MS experiments in public databases are represented as binary interactions according to the spokes model.

The literature quoted is biased towards the author's own contribution. For example, they are not the ones who first described the use of AP-MS for the charting of protein complexes. This was done first by Bertrand Seraphin in 1999! It would be great to see a more balanced (and fair) reference list.

We added the suggested reference on page 4 of our revised version to acknowledge the first tandem affinity purification applied for the characterization of yeast protein complexes.

Minor points:

- Supplementary Table 3 should also include the source or reference (PubMed ID, etc)

This information is included in Supplementary Table S3.

Reviewer #2:

The Hippo pathway plays an important and conserved role in controlling organ growth. This manuscript describes identification by mass spectrometry of a Hippo-pathway interactome, obtained by using several known components as baits. The authors also performed some limited functional validation of the ability of some of the interacting proteins they identified to modulate Hippo signaling when over-expressed in cultured cells, using a transcriptional reporter assay. The characterization of part of the interaction network under different conditions of cell attachment is also a nice addition. The manuscript does not provide compelling new insights into the pathway, but describes a screen that identifies candidate new players and interacting modules. I think the network of physical interactions identified by these studies is an interesting and potentially valuable resource for Hippo pathway research, and hence will be of general interest. In analyzing the interacting proteins obtained, the authors emphasize the expected interactions they identified along with the novel interactors. I would find it of interest if they could also comment on any known interactors of the baits they used that were not identified in their studies.

We have revised supplementary Table S3 to indicate interactions missed in our study and included a Venn diagram in the revised Supplementary Figure S2D to illustrate this more clearly (see comment above).

Reviewer #3:

This manuscript describes the analysis of the human Hippo growth regulatory pathway. The authors describe the proteomic analysis of this network and they obtain 480 protein-protein interactions between 270 network components. The authors use standard and established quantitative proteomic methods to analyze this network. The main biological aspects of the networks that the authors describe in part is the impact of interacting proteins on a reporter assay of the transcriptional activator YAP1 and differences in YAP1 interactions when cells are grown under adhesion or suspension conditions. The data in this manuscript is appropriately generated using standard approaches, and there is some follow up biology.

However, the lack of novel methods or mechanistic/functional insight into the network described results in this manuscript falling short of what is expected from a manuscript in Molecular Systems Biology. This is particularly evident if one compares this manuscript to the recently published MSB paper on the human HDAC protein interaction network from the Cristea group. In that paper, several other methods were used to validate the network and provide novel insights into the network. This included the use of imaging to determine colocalization of novel interacting partners, co-immunoprecipitation to validate novel interactions, and a clever use of SAINT scores and I-Dirt stability ratios to determine the relative interaction stability of HDAC-containing complexes.

This is the major issue with the current manuscript on the human Hippo pathway. Essentially, it provides a reported high confidence list of interactors, but there is no validation of these interactors with methods other than proteomic methods. Co-localization of components with each other co-immunoprecipitation studies are valuable to provide greater confidence in results to a wider audience of researchers rather than proteomics/systems biology researchers alone.

We addressed this point by testing a set of 35 interactions identified in our AP-MS experiments by co-immunoprecipitation and Western blotting. This involved the transfection of the corresponding epitope tagged versions of bait and prey proteins in HEK293 cells followed by immunoaffinity purification using anti HA antibodies and Western blotting against the V5 epitope present in the prey proteins. Similar to reports by others (Sowa et al. Cell, 2009) and our work (Varjosalo et al., Cell Rep. 2013) we could confirm 86% of the interactions found in AP-MS experiments. We describe the experimental validation rate now in the main text of our revised manuscript (page six) and have prepared a new supplementary Figure S1 that documents the experimental validation of protein interactions identified by AP-MS.

Also, there are no novel methods that are present in the current manuscript that would potentially increase the value of this manuscript.

One area of follow up biology that would need extensive additional work that would raise the interest of this manuscript relate to luciferase based assay regarding YAP1 activity stimulation. This data is in Figure 4B. Here the authors show that in this assay, PPIG, PPIA, and ASPP2 stimulate luciferase activity. There are a couple of issues with the way this experiment is described that first need to be clarified. Based on the figure legend and the manuscript it is unclear if these proteins are stimulating activity or maintaining activity.

The addition of YAP1 to the assay has the highest level of activity followed by PPIG, PPIA, and ASPP2. Are the authors assuming that in this assay endogenous YAP1 is present?

We thank the reviewer for pointing this out. The way we presented our results in previous Figure 4b was indeed not clear enough. In the revised version of Figure 4 we have changed the illustration on the experimental scheme and included additional luciferase reporter assays to address the questions of referee 3. We measured the effect of overexpression of proteins following transient transfection on transcriptional activation of TEAD promoters shown in Figure 4B in the presence of exogenous YAP1 expressed from a doxycycline inducible promoter stably integrated in a HEK293 cell line to enhance the sensitivity of our analysis. The additional transfection of YAP1 and MST1 was simply used as a positive control for the reporter assay (Supplementary Figure S5B). This is now clearly explained in the legends to Supplementary Figure 5 and in the main text pages 12 to 13..

Is YAP1 the only transcription factory that binds TEAD sites?

Having a completely in vitro way to test the effect of these proteins on YAP1 activity would benefit these results since it is not clear that these proteins are stimulating activity via YAP1. Perhaps another

control to add to this would be knocking down YAP1 in these cells and then seeing if PPIG, PPIA, and ASPP2 still stimulate activity.

YAP1 does not act as a DNA binding transcription factor that binds to TEAD sites directly. Instead, YAP1 acts as a transcriptional coactivator of TEAD DNA binding transcription factors. Following the reviewers suggestion we have performed additional reporter assays, which demonstrate that the observed PP1 mediated increase in TEAD reporter assay activity was indeed dependent on YAP1 protein levels and required TEAD binding. First, by applying siRNA against YAP1 we found that the observed PP1 mediated activation is dependent on YAP1 protein levels (Figure 4C). Second, the observed activation by PP1 was clearly dependent on the presence of TEAD transcription factor binding sites (Supplementary Figure S5C). These results are explained in the text on page 13. Taken together these results are consistent with a positive role of PP1 complexes in the activation of YAP1 dependent transcription in human cells.

In addition, do these three proteins localize with YAP1 in cells?

Here imaging studies would be valuable to co-localize these proteins in the nucleus, for example.

Finally, to provide additional data to support this potential mechanism, ChIP-Seq should be done with YAP1, PPIG, PPIA, and ASPP2 to see if they co-localize DNA in the cell. These datasets would greatly strengthen this manuscript

The presented AP-MS data together with existing literature information strongly suggest that PP1 as well as YAP1 form a number of concurrent complexes with other cellular proteins. These findings are reflected in both nuclear and cytoplasmic localization patterns reported in the literature for YAP1 as well as PP1, which make it rather difficult to conclusively support our functional and physical interactions by additional colocalization studies. We also have performed colocalization experiments using confocal immunofluorescence microscopy for YAP1, PP1a and ASPP2 in HEK293 cells. These results confirm cytoplasmic and nuclear localization for YAP1. Consistent with already published results, our experiments also revealed that the majority of ASPP2 protein is localized in the cytoplasm in HEK293 cells (Uhlmann-Schiffler et al, 2010, Oncogene; Wang Z. al., 2012, PNAS), whereas the bulk of PP1a is preferentially but not exclusively nuclear (Trinkle-Mulcahy L. et al., 2001, JCS). We conclude that the partly overlapping staining patterns likely reflect the localization of the entire pool of concurrent YAP1 and PP1 protein complexes in human cells but do not allow for a conclusive claim on the localization of YAP1-PP1 complexes.

The other area of biology that could be expanded greatly is the role of YAP1 interactions under adhesion and suspension growth. However, currently this data is problematic. The authors need to provide statistical significance for these results and provide the justification for this. Certainly the decrease in interaction of AMOT, AMOL1, and AMOL2 are dramatic. Following up on these results would also strengthen the manuscript. Here again, imaging and PCR would potentially be valuable. Does the localization of YAP1 and these three proteins change in the cells or is YAP1 somehow affecting the transcription levels of these three proteins? Here again, more mechanistic and functional information would be highly valuable to strengthen this manuscript.

We thank the reviewer for his suggestion to include statistical significance measures in the presentation of our quantitative MS data. Following the suggestion of referee 3 we repeated the quantitative AP-MS experiments on YAP1 complexes isolated from cells grown under adhesion or suspension conditions to represent the observed dynamics in biological triplicates in the revised version. The abundance profiles with the corresponding t-test statistics are now included in the revised Figure 6A.

To further support our model on the regulation of YAP1 complex formation by cell-cell contacts we performed an additional quantitative AP-MS experiment where we analyzed YAP1 protein complex formation in HEK293 cells grown under high and low density, since similar conditions have been previously reported to affect YAP1 localization and transcriptional activity in Eph4 and NIH3T3 cells (Varelas X. et al., 2011, Dev. Cell; Ota and Sasaki, 2008, Development). All quantitative AP-MS measurement were statistically analyzed using a t-test from biological triplicate experiments and statistical significance is indicated in the revised version of Figure 6B. Similar to the results obtained for suspension growth our new results revealed a decrease in YAP1 complex formation with cell polarity network components when cells are grown at low density. In addition we also performed in parallel transcription reporter assays and found enhanced transcriptional activity on TEAD luciferase reporter assays at low density. These new findings are now included in the revised version of Figure 6C and are discussed in the text (page 15-16). Taken together these results further support our initial claims on changes in YAP1 complex formation following changes in cell-cell contacts and in addition suggest that these changes may be linked to the control of YAP1 transcriptional activity.